# Depicting Fecal Microbiota Characteristic in Yak, Cattle, Yak-Cattle Hybrid and Tibetan Sheep in Different Eco-Regions of Qinghai-Tibetan Plateau

Xiaoqi Wang,[a,b,c] Zhichao Zhang,[a] Biao Li,[a] Wenjing Hao,[a] Weiwen Yin,[a] Sitong Ai,[a] Jing Han,[a] Rujing Wang,[b,c] Ziyuan Duan[a]

[a]Institute of Genetics and Developmental Biology, Chinese Academy of Sciences, Beijing, China
[b]Hefei Institutes of Physical Science, Chinese Academy of Sciences, Hefei, China
[c]University of Science and Technology of China, Hefei, China

**ABSTRACT** The gut microbiota is closely associated with the health and production performance of livestock. Partial studies on ruminant microbiota are already in progress in the Qinghai-Tibetan Plateau Area (QTPA) in China, but large-scale and representative profiles for the QTPA are still lacking. Here, 16S rRNA sequencing was used to analyze 340 samples from yak, cattle, yak-cattle hybrids, and Tibetan sheep, which lived in a shared environment from 4 eco-regions of the QTPA during the same season, and aimed to investigate the fecal microbiota community composition, diversity, and potential function. All samples were clustered into 2 enterotypes, which were derived from the genera *Ruminococcaceae UCG-005* and *Acinetobacter*, respectively. Environment, human activity, species, and parasitization all affected the fecal microbiota. By assessing the relationship between the fecal microbiota and the above variables, we identified a scattered pattern of fecal microbiota dissimilarity based more significantly on diet over other factors. Additionally, gastrointestinal nematode infection could reduce the capacity of the bacterial community for biosynthesis of other secondary metabolites, carbohydrate metabolism, and nucleotide metabolism. Ultimately, this study provided a fecal microbiota profile for ruminants living in 4 eco-regions of the QTPA and its potential future applications in developing animal husbandry regimes.

**IMPORTANCE** Cattle, yak, and sheep reside as the main ruminants distributed throughout most regions of Qinghai-Tibetan Plateau Area (QTPA) in China. However, there is a lack of large-scale research in the QTPA on their fecal microbiota, which can regulate and reflect host health as an internalized "microbial organ." Our study depicted the fecal microbiota community composition and diversity of yak, cattle, yak-cattle hybrids, and Tibetan sheep from 4 eco-regions of the QTPA. Additionally, our results demonstrated here that the ruminant samples could be clustered into 2 enterotypes and that diet outweighed other factors in shaping fecal microbiota in the QTPA. This study provided a basis for understanding the microbiota characteristic of ruminants and its possible applications for livestock production in the QTPA.

**KEYWORDS** fecal microbiota, enterotype, eco-region, 16S rRNA sequencing, machine learning, gastrointestinal nematode, ruminant

The Qinghai-Tibetan Plateau Area (QTPA) of China covers a region of 2,500,000 km² with an altitude of 3,000 to 5,000 m. In this vast territory, cattle, yak, and sheep are the main ruminant livestock distributed throughout most regions. Many studies have already demonstrated the importance of gastrointestinal tract bacteria which have close relations with the host health and production performance of economically important animals (1). It is clear that the study of the gastrointestinal tract microbiota and its potential functions can help improve livestock performance and immunity in the QTPA. Although a couple of studies

Address correspondence to Ziyuan Duan, zyduan@genetics.ac.cn.

The authors declare no conflict of interest.

have been conducted to characterize the gastrointestinal tract microbiota of ruminants in the QTPA, both of these either focused on the rumen bacterial community (2) or collected samples in only one or two regions (3, 4). The sampling coverage was not comprehensive due to the vast territory and diverse in the natural environment. A large-scale, standardized, coherent gastrointestinal tract microbiota description is still lacking.

Generally, species (5), environment (6), human activity (7), and parasitization (8) are considered important factors which can shift the diversity and composition of the gastrointestinal tract microbiota. Environment, particularly diet, has been regarded as the dominant factor in shaping the gastrointestinal tract microbiota in humans (6, 9, 10). However, some research has emphasized that host identity has a stronger effect than diet on forming the gut microbiota in mammals, despite the microbial community response to diet being more flexible (5, 11). It is known that gastrointestinal nematodes (GIN) are one of the most common causes of parasite infections in ruminants, and epidemiologic studies have indicated that GIN infection could lead to a pro-inflammatory response with a gut microbial species increase in sheep in the laboratory (12). However, the extent of the impact of GIN compared to other factors is still unclear. Concurrent with the object and study region expanding, the extent to which various factors shape microbiota composition trends is controversial and unclear, because study design and experimental scale could limit identification of the dominant factor (11). The QTPA, then, would be considered conducive to comparison as a big natural laboratory, since it has been divided into 10 eco-regions (13) with special vegetation, similar biological resources, and naturally distributed ruminants in different regions across long periods.

Specifically, we aimed to characterize the fecal microbiota composition and diversity of domestic ruminants (cattle, yak, yak-cattle hybrids, and Tibetan sheep) dwelling in 4 typical eco-regions with homologous longitudes but different latitudes in the QTPA using 16S rRNA sequencing. Owing to samples collected from multiple species or varieties in one habitat, and varied ecotypes living in different regions, we assessed the extent to which diet shapes the gut microbiota in ruminants. Furthermore, considering that GIN was a common parasitic disease and could also perturb the host microbiota, we also evaluated the influence of GIN infection on fecal microbiota diversity and potential function.

## RESULTS

**Parasitological survey.** According to the species and sampling locations, the samples were grouped into 10 groups: Yak-CK (yaks in Caka), Yak-DQ (yaks in Diqing), Yak-MK (yaks in Mangkang), Yak-TD (yaks in Tongde), Cattle-CK (cattle in Caka), Cattle-DQ (cattle in Diqing), YC-DQ (yak-cattle hybrids in Diqing), TS-CK (Tibetan Sheep in Caka), TS-TD (Tibetan Sheep in Tongde), and TS-MK (Tibetan Sheep in Mangkang). The detection of GIN parasites (Chromadorea class), as shown in Fig. 1, was determined by PCR amplification and internal transcribed spacer (ITS) region cloning of microbial rDNA in fecal samples.

The results showed that the rates of positive GIN infection were increasing in Tongde to Caka, Diqing, and Mangkang by 46.23%, 71.88%, 85.25%, and 89.61%, respectively. The infection rates in sheep were higher (Tibetan sheep, 81.25%) than those in *Bos* (yak, cattle, and yak-cattle hybrids, 63.68%). Based on GIN infections, samples were then divided into 4 groups: Bos-N (GIN-negative *Bos*), Bos-P (GIN-positive *Bos*), Sheep-N (GIN-negative sheep), and Sheep-P (GIN-positive sheep).

**Alpha and beta diversity.** Shannon diversity and Chao1 indices were employed to evaluate the diversity and richness of the bacterial community in feces, respectively. As shown in Fig. 2, wherever yak lived in different vegetation regions, the fecal microbial diversity was higher than that in cattle in the same habitat. For instance, in Diqing, Shannon indices in yaks was higher than those in cattle ($P < 0.05$, Fig. 2a) despite the fact that no differences in Chao1 index were detected among cattle, yak, and yak-cattle hybrids. There were significant differences ($P < 0.05$) in bacteria richness and diversity among the sheep groups from 3 different eco-regions (Fig. 2b). Among the yak population, the fecal microbiota in Yak-DQ and Yak-TD had a lower diversity than those in

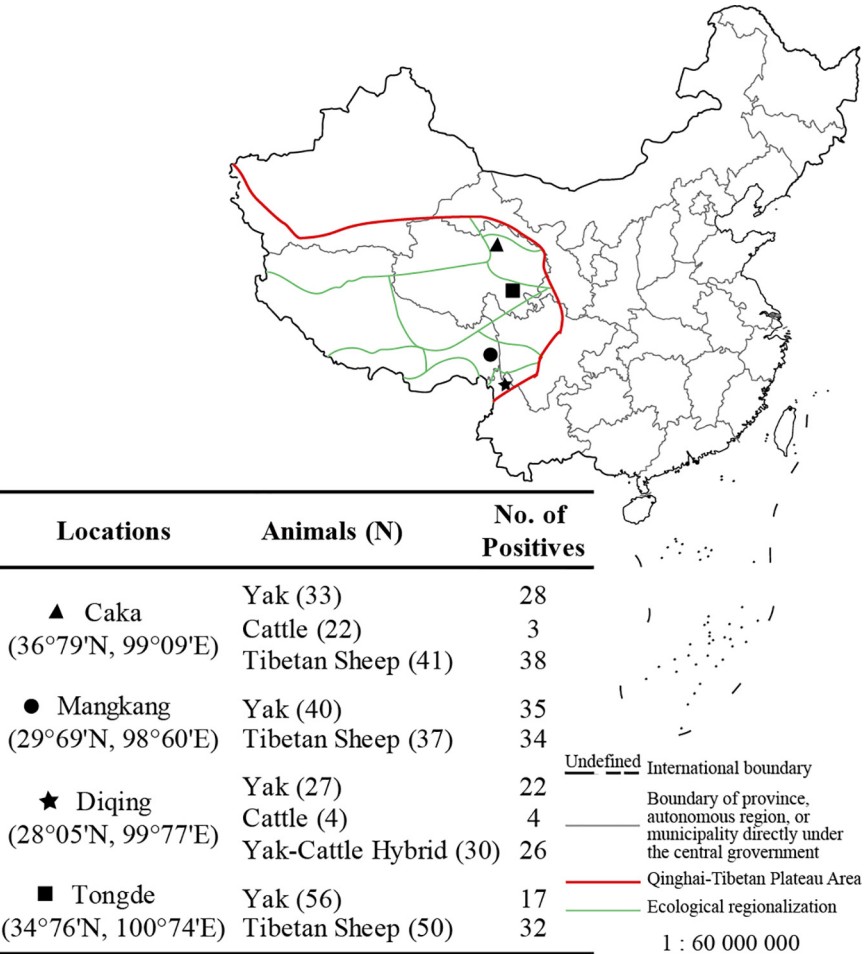

| Locations | Animals (N) | No. of Positives |
|---|---|---|
| ▲ Caka (36°79'N, 99°09'E) | Yak (33) | 28 |
| | Cattle (22) | 3 |
| | Tibetan Sheep (41) | 38 |
| ● Mangkang (29°69'N, 98°60'E) | Yak (40) | 35 |
| | Tibetan Sheep (37) | 34 |
| ★ Diqing (28°05'N, 99°77'E) | Yak (27) | 22 |
| | Cattle (4) | 4 |
| | Yak-Cattle Hybrid (30) | 26 |
| ■ Tongde (34°76'N, 100°74'E) | Yak (56) | 17 |
| | Tibetan Sheep (50) | 32 |

Undefined ——— International boundary

——— Boundary of province, autonomous region, or municipality directly under the central grovernment

——— Qinghai-Tibetan Plateau Area

——— Ecological regionalization

1 : 60 000 000

**FIG 1** Collection information and gastrointestinal nematode (GIN; class Chromadorea) infection among samples.

Yak-CK and Yak-MK ($P < 0.01$). In terms of GIN infection, alpha diversity indices increased when *Bos* and sheep were infected with GIN, and the difference between Sheep-N and Sheep-P was remarkable ($P < 0.05$, Fig. S1).

Weighted UniFrac distance-based beta diversity analysis revealed significant differences in the microbial community among the 10 groups ($P < 0.05$) and were performed by principal-coordinate analysis (PCoA). An analysis of similarities ($R > 0.5$, $P = 0.001$) demonstrated that the remarkable difference between groups was greater than that within groups. Similar to Tibetan sheep from 3 regions gathered separately (Fig. 2d), yak and cattle showed scattered patterns in fecal microbiota dissimilarity based on region rather than species (Fig. 2c). In addition, there was no difference between GIN-negative and positive groups, neither for *Bos* nor for sheep.

All samples were obtained from grazing ruminants in different ages and genders (see Table in Text S1 in the supplemental material). Therein, 4 populations of all ages and genders were included to reveal the effects on the diversity of bacterial communities (Fig. S2 in Text S1). However, age and gender had no significant effect on the beta diversity of the fecal microbiota in yaks, cattle, and Tibetan sheep.

**Taxonomic structure of the fecal bacteria.** Relative abundances of bacterial community composition at the phylum level across the 10 groups are shown in Fig. 3a, and significant differences in the bacterial phylum relative abundance for cattle, yak, yak-cattle hybrids, and Tibetan sheep are shown in Fig. S3 and S4 (see Text S1 in the supplemental material). Except for Cattle-DQ, the highest relative abundance was observed for phylum *Firmicutes*, followed by the phyla *Bacteroidetes*, *Proteobacteria*, and *Verrucomicrobia* in both

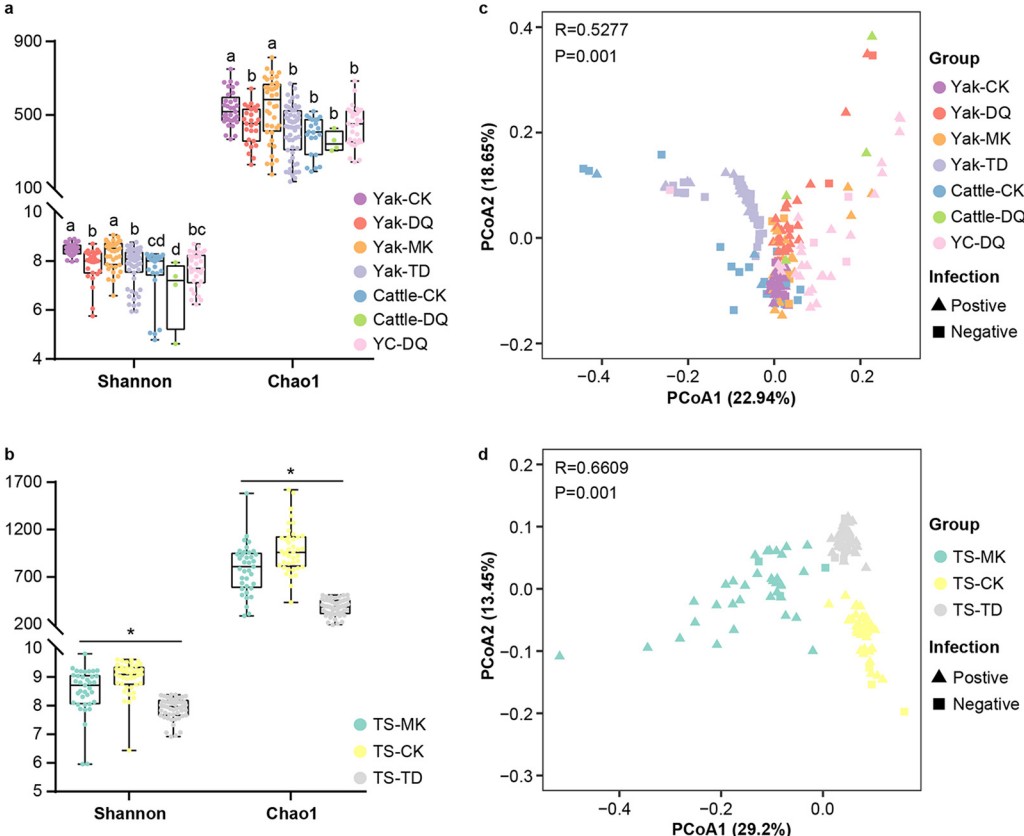

**FIG 2** Differences in fecal bacteria community composition among cattle, yak, yak-cattle hybrids, and Tibetan sheep. Alpha diversity based on Shannon and Chao1 indices of fecal bacteria among (a) Yak-CK, Yak-DQ, Yak-MK, Yak-TD, Cattle-CK, Cattle-DQ, and YC-DQ; (b) TS-CK, TS-MK, and TS-TD. Weighted Unifrac distance principal coordinate analysis (PCoA) plot of beta diversity measures of microbiota communities in fecal samples from (c) Yak-CK, Yak-DQ, Yak-MK, Yak-TD, Cattle-CK, Cattle-DQ, and YC-DQ; (d) TS-CK, TS-MK, and TS-TD. Yak-CK, Yak-DQ, Yak-MK, and Yak-TD: feces of yaks in Caka, Diqing, Mangkang, and Tongde, respectively. Cattle-CK and Cattle-DQ, feces of cattle in Caka and Diqing, respectively. YC-DQ, feces of yak-cattle hybrid in Diqing. TS-CK, TS-TD, and TS-MK, feces of Tibetan sheep in Caka, Tongde, and Mangkang, respectively. Significant differences are indicated by lowercase letters (analysis of variance [ANOVA]); *t* test was used for statistical analysis; *, $P < 0.05$.

groups. *Proteobacteria* had higher relative abundance in samples collected from Diqing (Yak-DQ, Cattle-DQ and YC-DQ) compared with that in samples from Mangkang (Yak-MK), Caka (Yak-CK and Cattle-CK), and Tongde (Yak-TD); whereas the relative abundance of *Verrucomicrobia* was lowest in samples from Diqing (Yak-DQ, Cattle-DQ and YC-DQ). Samples collected from Tongde (Yak-TD and TS-TD) had the highest *Firmicutes* abundance compared with that in other regions ($P < 0.05$), while the relative abundance of *Verrucomicrobia* was higher in samples collected from Caka (Yak-CK, Cattle-CK and TS-CK) than that in other regions ($P < 0.05$). In terms of species and variety, the relative abundance of *Firmicutes* was higher in yaks than in cattle or yak-cattle hybrids.

Among the GIN-positive and -negative groups, the top 4 abundant phyla exhibited the same variety. The relative abundances of *Firmicutes* and *Verrucomicrobia* were higher in GIN-negative samples both in *Bos* and sheep, but the reverse was true for *Bacteroidetes* and *Proteobacteria*. Interestingly, phylum *Chlamydiae* was detected in the YC-DQ and TS-MK groups (Fig. 3a), while it was merely present in GIN-positive samples from these 2 groups (Fig. S5a in Text S1).

Furthermore, taxonomy was assigned to 231 bacterial genera in all groups. Fig. 3b and 3c show the top 15% of bacterial genera ranked by relative abundance. *Ruminococcaceae UCG-005*, *Rikenellaceae RC9 gut group*, and *Ruminococcaceae UCG-010*, with great different abundances among groups, were the dominant genera both in Tibetan sheep and in yak, cattle, and yak-cattle hybrids. Several genera displayed strong preferences for species or region,

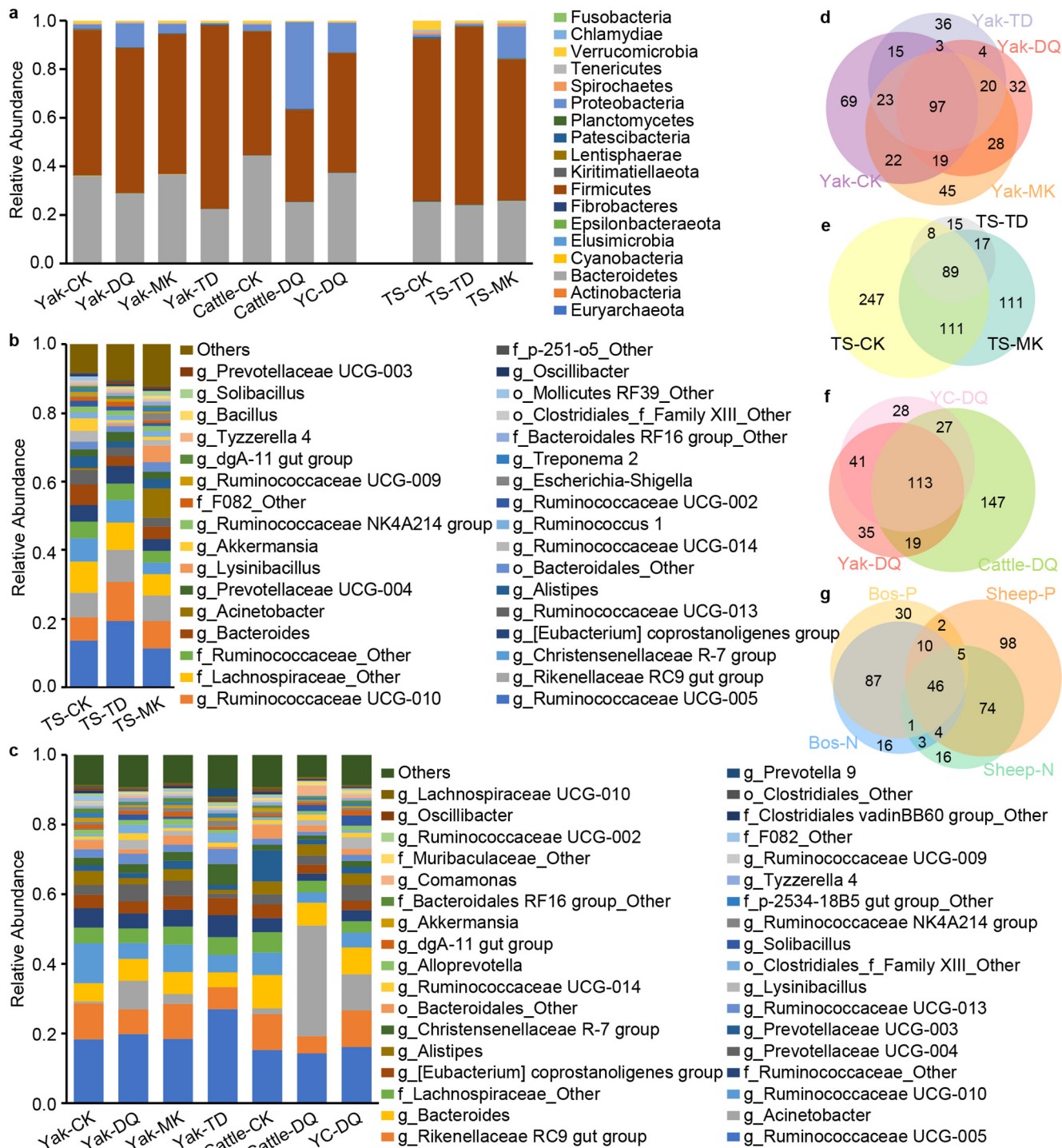

**FIG 3** Bacterial taxonomic composition. Bacterial taxonomic composition at (a) phylum level in all fecal samples, (b) genus level in Tibetan sheep, and (c) genus level in cattle, yak, and yak-cattle hybrids. Venn diagrams indicate the numbers of ASV among (d) yaks from 4 areas; (e) Tibetan sheep from 3 areas; (f) cattle, yak, and yak-cattle hybrids in Diqing; and (g) GIN-infected and uninfected bovines and sheep, respectively. All amplicon sequence variations (ASVs) existed as ≥ 50% of each group population.

such as *Ruminococcaceae UCG-005*, *Christensenellaceae R-7 group*, *Alloprevotella*, and *Ruminococcaceae UCG-013*, with higher abundances in yaks than in cattle or yak-cattle hybrids ($P < 0.05$); *Acinetobacter* had higher abundance while *Ruminococcaceae UCG-010* and [*Eubacterium*] *coprostanoligenes group* had lower abundances (yaks and cattle) in Diqing than in other regions ($P < 0.05$). Similarly, *Ruminococcaceae UCG-005* and [*Eubacterium*]

*coprostanoligenes group* in yak and Tibetan sheep samples collected from Tongde were at the highest abundances compared to those from other regions ($P < 0.05$). Among the highly abundant genera, the relative abundances of *Ruminococcaceae UCG-005*, *Bacteroides*, [*Eubacterium*] *coprostanoligenes group*, *Christensenellaceae R-7 group*, and *Prevotellaceae UCG-003* decreased, while *Rikenellaceae RC9 gut group* and *Alistipes* increased in both *Bos* and sheep samples (Fig. S5b in Text S1) when they were GIN-infected.

In this study, the amplicon sequence variants (ASVs) defined as at least 50% of each population group were regarded as "core ASVs." Hence, a total of 97 shared ASVs were identified from 4 regions in yaks (Fig. 3d), while less than 113 ASVs were detected among yak, cattle, and yak-cattle hybrids in Diqing (Fig. 3f). Likewise, there were only 89 shared ASVs among the Tibetan sheep groups (Fig. 3e). The fecal samples from *Bos* and sheep infected with GIN only shared 2 core ASVs, which were closely related to GIN-positive samples (Fig. 3g). In addition, one of the 2 ASVs was taxonomically assigned to genus *Acinetobacter*.

**Fecal microbiota characteristics and functional potential.** To screen the discriminant genera of fecal microbiota among cattle, yak, yak-cattle hybrids, and Tibetan sheep in different regions, we used a machine learning method, random forest (RF) classifier. We defined the 30 bacterial genera as the optimal taxa of yaks from 4 eco-regions (Fig. 4a) and Tibetan sheep from 3 eco-regions (Fig. 4b), respectively, according to the 10-fold cross-validation. Hence, the RF models were established at the genus level, showing 97.3% average accuracy for identifying biomarkers among yaks and nearly 100% accuracy for screening biomarkers among Tibetan sheep. Excluding unclassified genera, *Lysinibacillus* and *Ruminococcaceae UCG-002*, and *Acinetobacter* and *Christensenellaceae R-7 group*, are the top 2 biomarkers in terms of bacterial genera impact in yaks and Tibetan sheep, respectively.

Predictive function demonstrated a region-based pattern of potential function dissimilarity among 4 yak groups and 3 Tibetan sheep groups in the fecal microbiota (Fig. 4c). Similar to the region preference of some genera, the potential functions of microbiota communities also showed environment preference. In yaks and Tibetan sheep, amino acid, carbohydrate, nucleotide, terpenoid and polyketide metabolism were more strongly enriched, whereas the biodegradation and metabolism of xenobiotics was less abundant in Tongde than in other regions. Terpenoid and polyketide metabolism had the highest abundances in yaks and Tibetan sheep sampled from Tongde, followed by Caka, Mangkang, and Diqing. Compared to those in other regions, yaks and Tibetan sheep in Caka were associated with lower potential lipid metabolism. In addition, there were no remarkable differences among yak, cattle, and yak-cattle hybrids in Diqing (data not shown).

Additionally, to explore whether fecal microbiota can serve as biomarkers to distinguish animals from these 4 eco-regions of the plateau as GIN-positive or GIN-negative, another RF model (with a false discovery rate of 23.73%) was established at the genus level. Thirty differentially abundant genera as the optimal taxa were screened from samples with/without GIN infection (Fig. S6 in Text S1). According to RF prediction, *Ruminococcus 1*, *Ruminiclostridium 6*, and *Alloprevotella* were the most discriminating markers. Although there were no significant differences in microbial diversity, the error rate was high between GIN-positive and -negative samples, and 8 metabolic functions with statistically significant difference were found between them (Fig. 4c). From this, 4 categories were discovered in both *Bos* and sheep. Interestingly, the GIN-infected and uninfected samples had roughly the same trend in both *Bos* and sheep. The GIN-negative groups (Bos-N and Sheep-N) showed increased capacity for biosynthesis of other secondary metabolites, carbohydrate metabolism, and nucleotide metabolism; and decreased capacity for xenobiotic biodegradation and metabolism, compared to the GIN-positive groups (Bos-P and Sheep-P).

**Enterotype distribution profiling of sampled animals.** According to the assessment method of Arumugam et al. (14) and the Calinski-Harabasz (CH) index (Fig. S7 in Text S1), 340 samples from yak, cattle, yak-cattle hybrids, and Tibetan sheep from 4 eco-regions in the QTPA were clustered into 2 enterotypes (Fig. 5a) and their representative genera were assessed based on the dominating genus in each one (Fig. 5b). The abundance of *Ruminococcaceae UCG-005* (belonging to phylum *Firmicutes*) was much higher than

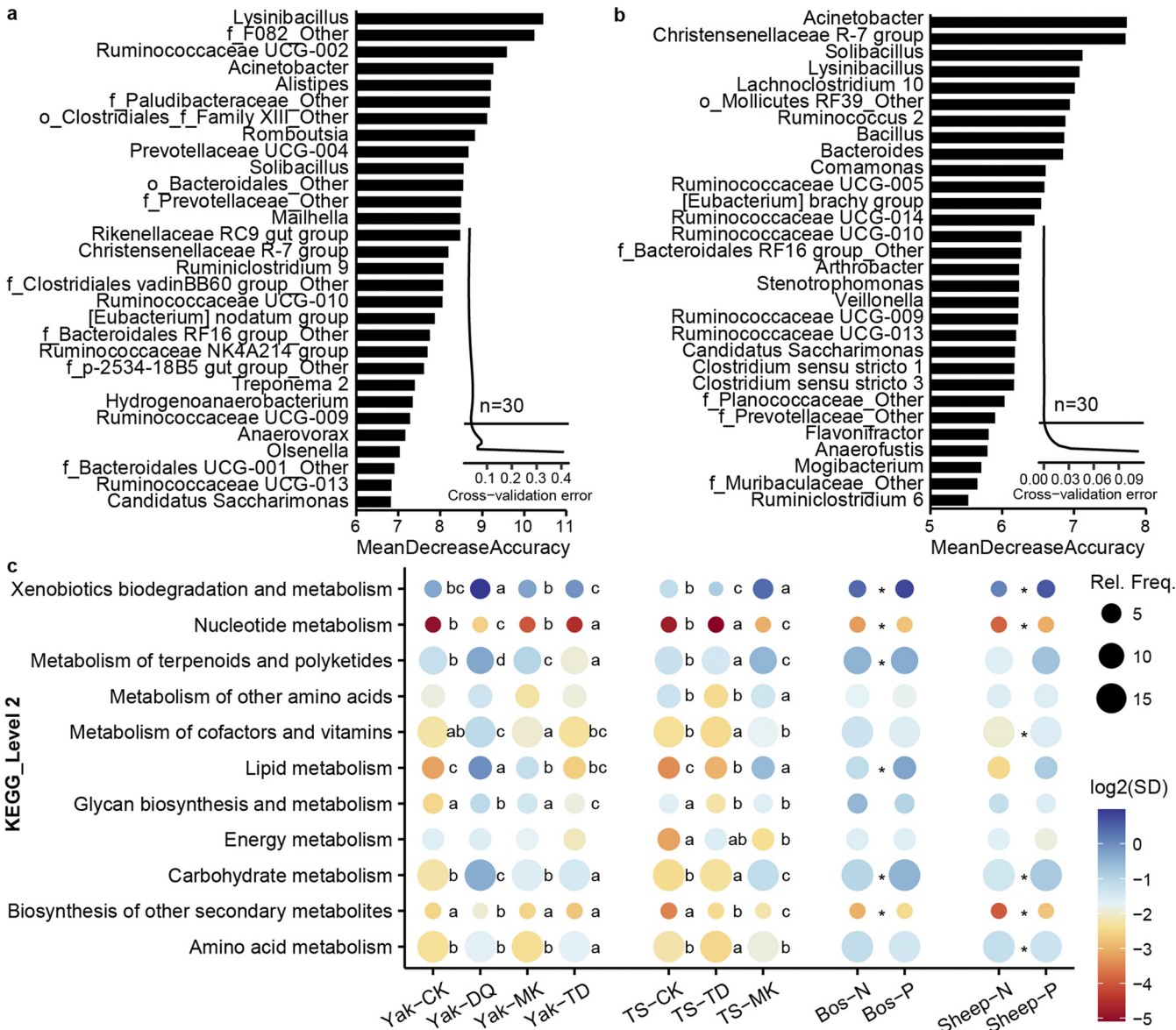

**FIG 4** Microbial biomarkers screened by random forest (RF) method at genus level and potential functions analyzed using PICRUSt2. Bacterial genera were identified by RF classification of relative abundances in fecal microbiota among (a) yaks from Caka, Diqing, Tongde, and Mangkang; and (b) Tibetan sheep in Caka, Tongde, and Mangkang. Curves represent the 10-fold cross-validation to determine the credible number of genera. Biomarker taxa are ranked in descending order of importance to accuracy. (c) Differential metabolism pathways among yak, Tibetan sheep, and GIN-infection groups. Significant differences are marked by lowercase letters among Yak-CK, Yak-DQ, Yak-MK and Yak-TD; and among TS-CK, TS-TD, and TS-MK (ANOVA); a *t* test was used for statistical analysis between Bos-N and Bos-P and between Sheep-N and Sheep-P. *, $P < 0.05$. The $P$ values above were corrected by Benjamini-Hochberg false-discovery rate (FDR). Rel.Freq., relative frequency; SD, standard deviation; Yak-CK, Yak-DQ, Yak-MK and Yak-TD, feces of yaks in Caka, Diqing, Mangkang, and Tongde, respectively; TS-CK, TS-TD and TS-MK, feces of Tibetan Sheep in Caka, Tongde, and Mangkang, respectively; Bos-N and Bos-P, feces of *Bos* without/with GIN infection; Sheep-N and Sheep-P, feces of sheep without/with GIN infection.

that of other genera in enterotype *Ruminococcaceae UCG-005* (E1), while *Acinetobacter* (belonging to phylum *Proteobacteria*) was the most abundant genus, with *Ruminococcaceae UCG-005* a close second, in enterotype *Acinetobacter* (E2). Density curves show the distributions of *Ruminococcaceae UCG-005* and *Acinetobacter* in E1 and E2 (Fig. S8 in Text S1). As shown in Fig. 5c, up to 85.9% samples belonged to E1, which existed in all groups, but E2 has rarely been reported. It was almost detected in samples from Diqing and Mangkang and in only one sample from Caka. Although E2 existed in both GIN-positive and -negative groups, many more of the GIN-infected samples than GIN-noninfected samples belonged to E2.

Due to a better understanding of correlations in co-occurring genera, we performed a Spearman's correlation coefficient based on genus abundances (Fig. 5d). With a correlation

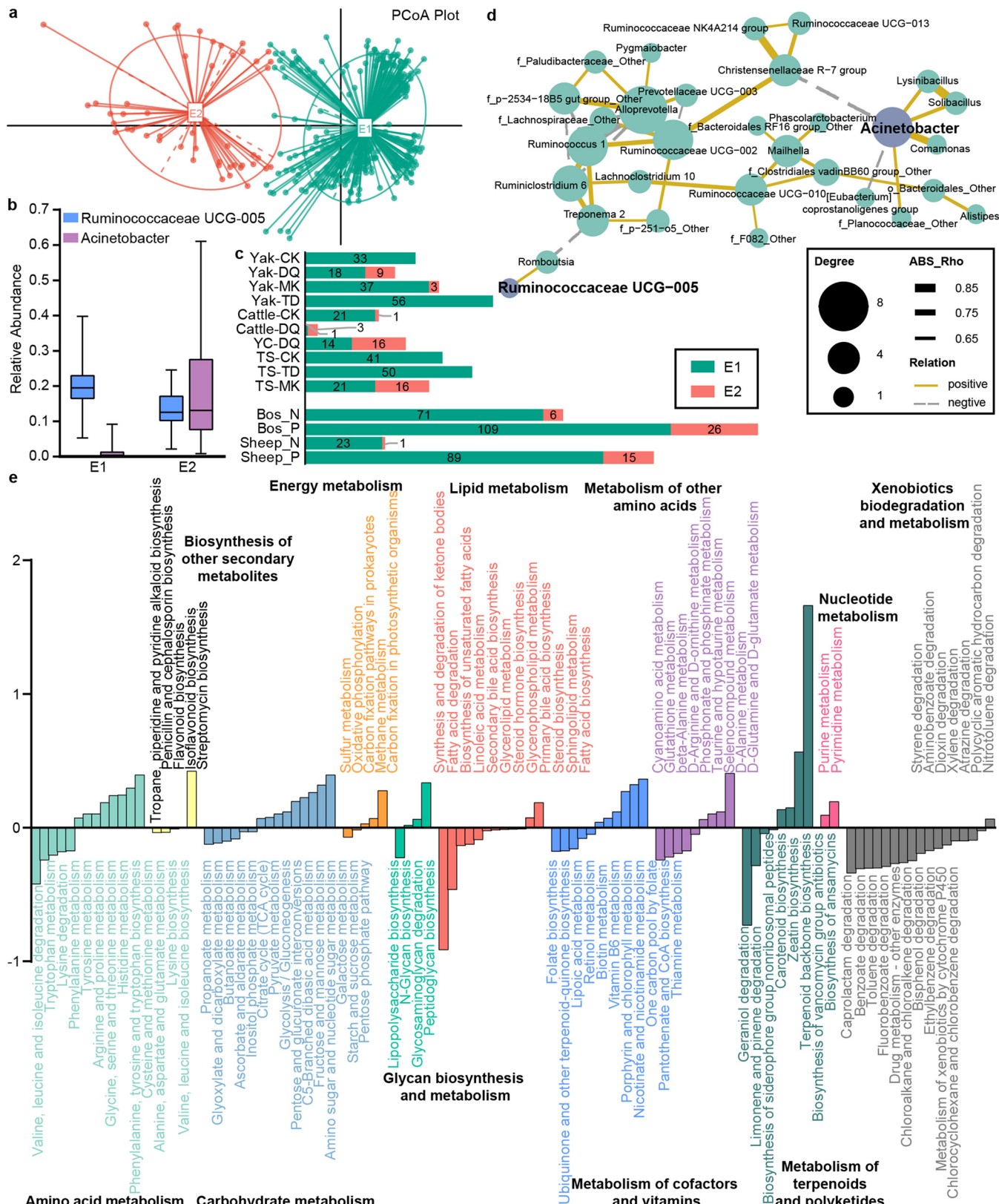

**FIG 5** Analysis of enterotypes across cattle, yak, yak-cattle hybrids, and Tibetan sheep. (a) PCoA plot showing that all samples were assigned to 2 enterotypes based on Jensen-Shannon divergence distance. (b) Relative abundances of the representative genera (*Ruminococcaceae UCG-005* and *Acinetobacter*) of each enterotype. (c) Bar chart shows numbers of samples belonging to enterotypes of each group. (d) Spearman's correlation coefficients with $R > 0.6$ and $P < 0.05$ are shown. Each node represents a genus. Solid and dotted lines indicate positive and negative correlations, respectively. (e) The functional potentials of bacteria communities between E1 and E2 were predicted based on KEGG using PICRUSt2. Differential metabolism pathways (KEGG_3, $P < 0.05$) were showed in different colors. The histogram represents the values of E1 minus E2. A $t$ test was used for statistical analysis and $P$ values were corrected by Benjamini-Hochberg FDR.

coefficient (|R|) over 0.6 and a *P* value of <0.05, *Comamonas*, *Lysinibacillus*, *Solibacillus*, and f_*Planococcaceae_Other* were positively correlated with *Acinetobacter*, while *Christensenellaceae R-7 group* and [*Eubacterium*] *coprostanoligenes group* were negatively correlated with *Acinetobacter*. Additionally, *Ruminococcaceae UCG-005* was directly proportional to *Romboutsia*.

Generally, enterotype can lead to significant metabolism functional variation. To explore the bacterial community functions of E1 and E2, which were established by different driving genera and their co-occurring genera, we used PICRUSt2 and STAMP2.1.3 to carry out analysis. Fig. 5e shows the significant differences in metabolism-related functions between E1 and E2 (*P* < 0.05). Considering that *Ruminococcaceae UCG-005* was positively correlated with short-chain fatty acid production and *Acinetobacter* is a glucose-nonfermentative bacteria with a drug-resistant character, we specifically focused on lipid metabolism (12 categories), carbohydrate metabolism (15 categories), terpenoid and polyketide metabolism (8 categories), and xenobiotic biodegradation and metabolism (17 categories). Obviously, E1 was associated with a higher potential of fatty acid biosynthesis; however, the bacteria communities of E2 were enriched in propanoate, glyoxylate and dicarboxylate, butanoate, ascorbate and aldarate, inositol phosphate metabolism, the tricarboxylic acid (TCA) cycle, and the overwhelming majority of xenobiotic biodegradation pathways.

## DISCUSSION

This study systematically evaluated the differences in the gut microbiota community composition and diversity of yak, cattle, yak-cattle hybrids, and Tibetan sheep from the same or different regions located in homologous longitudes but in different latitudes in the QTPA in China. Due to specific factors of the QTPA, such as low average annual temperature, vagaries of climate and vegetation, and altitude differences, comprehensive reference data for gut microbiota in ruminants in the QTPA are very limited. This research would sketch the contours of fecal microbiota characteristics in the main ruminants along similar longitudes with extreme vegetation situations (15, 16).

Generally, enterotypes can be used for potential disease prediction and medical applications, and 3 enterotypes (*Bacteroides*, *Prevotella*, and *Ruminococcus* enterotype) were defined in humans (14, 17). There are many variables in switching enterotype, such as diet, age, season and so on (17), but the variable is still a research blank in shifting enterotype of sheep and cattle in the QTPA. Recently, a research team has shown that the gut microbiota of neighboring yaks was partitioned into the *Akkermansia* and uncultured *Eubacterium WCHB1-41* enterotypes, the *Ruminococcaceae_UCG-005* enterotype, and the *Ruminococcaceae_UCG-010* enterotype, and the dynamics of enterotype and season (diet separation) (9). Inconsistent with this, we performed enterotype analysis based on different species (yak, cattle, and sheep) collected from 4 eco-regions in the QTPA, defined as *Ruminococcaceae UCG-005* enterotype (E1) and *Acinetobacter* enterotype (E2). As previous reported, E1 preferred a warm climate and a diet with high protein and low fiber (9). The high frequency of E1 samples and high *Ruminococcaceae UCG-005* abundance in both E1 and E2 might be derived from summer sampling. A study on African buffalo identified 2 putative enterotypes: *Ruminococcaceae_UCG-005* enterotype and *Solibacillus* enterotype (18). In this study, the *Solibacillus* enterotype was closely related to diet and the driving genus, *Solibacillus*, was a co-occurring genus and positively associated with *Acinetobacter* in E2 in this study. E2 mainly existed in Diqing and Mangkang, which are close in geographic location and have higher population density than other regions according to a demographics data set (19). Frequent human activities and a wide application of chemicals has shifted relative bacteria abundance and functions (7), subsequently forming E2. Human activities are closely related to pollutants, drugs, chemicals, and antibiotics, which has disturbed the microbiota of the environment and organisms (7). For instance, the increased *Lysinibacillus* was highly resistant to heavy metal pollution in farmland systems (20), and the abundance of *Acinetobacter* rises in pesticide-exposed areas (21). Another co-occurring genus, *Comamonas*, has also been confirmed to be

highly correlated with sulfur metabolism, nitrogen metabolism, and the TCA cycle (22), in accordance with functional predictions. This study provides microbiological insights on how the enterotype reflects the local vegetational and ecological information, and that the driving genera and their co-occurring genera in enterotypes play critical roles in mediating nutritional and energy metabolism in ruminants in the QTPA.

In this study, environment was the major determinant over other factors in shaping fecal microbiota. Specifically, the fecal microbiota of ruminants was more similar to the closely related varieties/species living in one habitat than to the same species living elsewhere, such as there were more shared ASVs among *Bos* in Diqing than among yaks from different regions. As well as human activities, altitude, temperature, humidity, and diet, which are all environmental factors, could determine the diversity and composition of the gastrointestinal tract microbiota (23–25). For comprehensive consideration of sampling region information, fecal microbiota composition and diversity, bacterial community potential functions, and enterotype analysis, we would have preferred that remarkable differences were predominantly attributable to diet. A study on rumen and camelid foregut microbiota of 742 samples from diverse species and multiple countries also showed a that core bacterial community existed and diet had more influence than host species (26).

According to the ecological region division in the QTPA (13), Caka (3000 m), Diqing (3300 m), Mangkang (4,000 m), and Tongde (3,400 m) belong to 4 eco-regions with different vegetative cover, which can be represented by the Normalized Difference Vegetation Index (NDVI) (27) to monitor general vegetation change (28). Vegetation plays a key role in ecosystems and can comprehensively reflect regional climate conditions. Here, the NDVI of Caka, Mangkang, Diqing, and Tongde were about 0.20, 0.70, 0.83, and 0.80, respectively, and aboveground biomass was 44.38, 71.98, 141.34, and 115.63 $g/m^2$, respectively, as shown by an NDVI distribution map and biomass data set (15, 16). In agreement with vegetation distribution, phylum *Verrucomicrobia* was involved in carbohydrate metabolism, and the main genus, *Akkermansia muciniphila*, which can utilize host mucin as a carbon source in case of insufficient food (29), was significantly enriched in samples from Caka, which had the lowest NDVI and biomass, and less so in Diqing which had the highest NDVI and biomass. A recent report systematically reviewed the correlation of gut microbiota composition with obesity in humans and indicated that *Proteobacteria* had a positive correlation and *Verrucomicrobia* a negative correlation with obesity (30). This evidence suggests that the highest *Proteobacteria* and lowest *Verrucomicrobia* abundances in samples from Diqing owed to sufficient food (abundant grass resources). In general, the regional preference of the microbiota potential functions was in accordance with the driving genus; for example, animals in Tongde had the highest abundance of *Ruminococcaceae UCG-005*, which is specifically associated with acetate level (31), predictive for the enrichment of carbohydrate metabolism. Furthermore, potential horizontal transmission with environmental microbe among animals in one habitat (32), and the social behaviors of animals (33) may increase the core ASVs and enhance the structure of sympatric fecal microbiota, either.

Host species also had an effect on the fecal microbiota diversity. For instance, the specific performance of the Shannon diversity index of Tibetan sheep was the highest, followed that of by yak, yak-cattle hybrids, and cattle in every eco-region. It is probable that some gastrointestinal tract bacteria, such as species-unique bacteria, were influenced by host genetics. Actually, a previous study reported 19 single nucleotide polymorphisms of 709 beef cattle which were related with rumen microbial taxa (34). Additionally, the increasing diversity of gut bacteria in yaks evolved efficiently utilized various substances to adapt to food shortage (35), compared with that in cattle. However, species was inconsistent with environment and parasitic infection, and had little influence on shifts in enterotype. Nutritional and metabolic disorders of the host were closely tied to increases in *Proteobacteria* abundance, which was generally associated with lipopolysaccharide biosynthesis and dysbiosis (36, 37). In this study, GIN may have cause *Proteobacteria* to increase remarkably and then alter the metabolism in GIN-positive samples. Additionally, the observed repression of carbohydrate transport

and metabolism during GIN infection in *Bos* and sheep was consistent with the findings of a report on piglets in Beltsville, owing to *Trichuris suis* (belonging to GIN) reducing bacteria which could utilize carbohydrates (38); however, the opposite result was observed in a child infected with *Strongyloides stercoralis* (belonging to GIN) in Rome (39). Apparently, the different species and diets led to exactly opposite results. Thus, under the influence of diet and host factors, assessment of the relationship between GIN infection and the fecal microbiota was limited and unclear.

Ultimately, due to ruminants in different ages requiring different nutrition, the rumen microbiota changes along with yak and Tibetan sheep growth (40, 41). However, there was also a report that the microbial community composition of the rumen was similar in 2-year-old compared to 3-year-old cattle-yak hybrids (42). Besides this, the gender of yaks exerted a weak effect on the rumen microbiota (43). In this study, we detected no significant effects of age and gender on the fecal bacteria communities in yak, cattle, and Tibetan sheep, although dynamic changes in gut microbial community based on age-dependent effects were observed in yaks of 3 age stages (1-, 5-, and 12-year-old) (44). Here, in contrast to the influence of diet and host factors on shifting the fecal microbiota, those of age and gender were comparatively smaller.

**Conclusions.** Being obvious climate and altitude reasons, wide sampling in the QTPA was very difficult; even so, we still collected as many samples in the same season as possible for this study to minimize the artificial factors confounding sample reliability. This study reveals the characteristics of the fecal microbiota in yak, cattle, yak-cattle hybrids, and Tibetan sheep in different eco-regions of the QTPA and types all samples into 2 enterotypes: *Ruminococcaceae UCG-005* enterotype (E1) and *Acinetobacter* enterotype (E2). In addition, E2 was first defined in studies of intestinal bacteria. Results also show a scattered pattern of fecal microbiota dissimilarity based on diet rather than host factor among ruminants in the QTPA. The consequences in the study are crucial to establishing the spectrum of fecal microbiota in ruminants at medium altitudes in the QTPA and providing potential future applications in developing animal husbandry regimes. As the importance of area-specific factors has been identified in this study, more samples from varied eco-regions are needed to collect to perfect the spectrum of the fecal microbiota of ruminants.

## MATERIALS AND METHODS

**Sample collection.** Feces were sampled from domestic animals (cattle, yak, yak-cattle hybrids, and Tibetan sheep) dwelling and naturally distributed in Caka (3,000 m), Mangkang (4,000 m), Diqing (3,300 m) and Tongde (3,400 m) in the QTPA in China, respectively, assigned to 4 typical eco-regions with differences in vegetation index (Fig. 1) (13). Sample information (gender and age) is shown in Table S1 (see Text S1 in the supplemental material). The sampling season was from June to July, during the summer half of the year in the QTPA. Yak-cattle hybrids were the F1 generation of male yak and female cattle. All animals were raised free-range and by grazing natural vegetation. Fecal samples were immediately picked up when the animals defecated, then collected by sterile tubes and cryopreserved immediately in liquid nitrogen. These were then transported to the laboratory and stored at −80℃.

**DNA isolation, parasite detection, and 16S rRNA sequencing.** DNA was isolated using a TIANamp stool DNA kit (Tiangen, Beijing, China) according to the manufacturer's instructions. GIN were detected by PCR analysis based on targeting internal transcribed spacer regions of ribosomal DNA (45), and the positive PCR products were bi-directionally sequenced to identify them (BGI, Beijing, China). The V3 to V4 regions of bacterial 16S rRNA were amplified using the 341F/805R primer set, and PCR conditions followed protocols (46, 47). The PCR products were sequenced on an Illumina HiSeq2500 platform (Health Genomics Bioinformatics Technology Co., Ltd., Beijing, China).

**Bioinformatics and statistical analysis.** The paired-end 16S rRNA sequencing data were joined and quality-filtered using the FLASH method (48); all sequences were analyzed using Quantitative Insights Into Microbial Ecology 2 (QIIME2) according to its tutorial (49). Further noisy-sequences filtering, error correction, and chimeric sequence and singleton removal were performed using DADA2 (50). The remaining tags were clustered into amplicon sequence variants with a cutoff of 99% similarity. ASVs were taxonomically assigned to the Silva database (release 138) and the taxa table was generated by the classify-sklearn taxonomy classifier in QIIME2. Next, alpha diversity was analyzed using Shannon and Chao1 indices, while beta diversity was estimated basing on weighted UniFrac distance and shown in a PCoA plot. Analysis of similarities in the vegan package of R4.0.3 with 999 permutations was used to detect differences within and between groups.

Microbial function analysis was performed using PICRUSt2 (51) based on ASVs clustered from 16S rRNA sequencing data, then metabolic predictions were identified from Kyoto Encyclopedia of Genes and Genomes database. Differences in predicted results were processed using analysis of variance (ANOVA) (among multiple groups) or Welch's $t$ test (between 2 groups) with the Benjamini–Hochberg false-discovery rate (FDR) correction in STAMP v2.1.3 (52). Differences were presented as significant at $P < 0.05$ and highly significant at $P < 0.01$.

The important bacterial taxa among groups were screened by the RandomForest package of R4.0.3 based on the relative abundances of bacterial taxa at the genus level. Scientific graphs were generated using the ggplot2 package in R4.0.3. Enterotype analyses were based on relative genus abundance using the Jensen-Shannon divergence (JSD) distance and the partitioning around medoids (PAM) clustering algorithm (14). The Calinski-Harabasz index was employed to decide an optimal cluster number, and this clustering was calculated by clusterSim and the cluster package of R4.0.3. Spearman's correlation analysis was carried to explore relationships between the predominant genera and other genera with a correlation coefficient (R) of $\geq 0.6$ or $\leq -0.6$ and a $P$ value of $<0.05$.

**Data availability.** The 16S rRNA sequence data reported in this paper have been deposited in the Genome Sequence Archive (National Genomics Data Center, China National Center for Bioinformation/Beijing Institute of Genomics, Chinese Academy of Sciences) under accession number CRA005287, which is publicly accessible at https://ngdc.cncb.ac.cn/.

## SUPPLEMENTAL MATERIAL

Supplemental material is available online only.
**SUPPLEMENTAL FILE 1**, PDF file, 0.9 MB.

## ACKNOWLEDGMENTS

We thank Zhao Kai (Northwest Institute of Plateau Biology, Chinese Academy of Sciences) for helping collect samples on the plateau. This work was supported by the Strategic Priority Research Program of the Chinese Academy of Sciences under grant no. XDA2004010305, and the Key Projects of Chinese Academy of Sciences under grant no. KFZD-SW-219.

X-W: Methodology, investigation, resources, validation, formal analysis, data curation, and draft writing. Z.Z.: Investigation and resources. B.L., W.H., W.Y., S.A., and J.H.: Resources. R.W.: Formal analysis. Z.D.: Conception and supervision of experiment, manuscript revision.

We declare we have no conflicts of interest.

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
