## [Reviewer comments · Microbiology Spectrum]

Microbiology Spectrum

Depicting Fecal Microbiota Characteristic in Yak, Cattle, Yak-Cattle Hybrid and Tibetan Sheep in Different Eco-Regions of Qinghai-Tibetan Plateau

Xiaoqi Wang, Zhichao Zhang, Biao Li, Wenjing Hao, Weiwen Yin, Sitong Ai, Jing Han, Ruijing Wang, and Ziyuan Duan

Corresponding Author(s): Ziyuan Duan, Institute of Genetics and Developmental Biology, Chinese Academy of Sciences

Review Timeline:

Submission Date:	January 4, 2022
Editorial Decision:	February 9, 2022
Revision Received:	May 8, 2022
Editorial Decision:	May 22, 2022
Revision Received:	June 3, 2022
Accepted:	June 8, 2022

Editor: Jing Han

Reviewer(s): Disclosure of reviewer identity is with reference to reviewer comments included in decision letter(s). The following individuals involved in review of your submission have agreed to reveal their identity: Bing Wang (Reviewer #2)

Transaction Report:

DOI: <https://doi.org/10.1128/spectrum.00021-22>

February 9, 2022

Prof. Ziyuan Duan
Institute of Genetics and Developmental Biology, Chinese Academy of Sciences
Beijing
China

Re: Spectrum00021-22 (Depicting Fecal Microbiota Characteristic in Yak, Cattle, Yak-Cattle Hybrid and Tibetan Sheep in Different Eco-Regions of Qinghai-Tibetan Plateau)

Dear Prof. Ziyuan Duan:

Link Not Available

Sincerely,

Jing Han

Journals Department
Reviewer comments:

Reviewer #1 (Comments for the Author):

Ms. ID Spectrum00021-22: "Depicting Fecal Microbiota Characteristic in Yak, Cattle, Yak-Cattle Hybrid and Tibetan Sheep in Different Eco-Regions of Qinghai-Tibetan Plateau"

General comments:

The aim of this study was to investigate the fecal microbiota community composition, diversity and potential function based on

16S rRNA sequencing to analyze 340 samples from yak, cattle, yak-cattle hybrid and Tibetan sheep living in four regions of Qinghai-Tibetan Plateau.

Specific comments:

Introduction:

Lines 60-69: The context of the study is described but the problem is not. What is the problem that justifies analyzing the fecal microbiota of four ruminants? Is it the presence of parasites, is it based on geographical differences? Without a clearly stated hypothesis or research question, this study is only descriptive although the number of fecal samples is high. Please identify the problem.

Lines 70-71: Why did you also study the fecal microbiota of Tibetan sheep? Besides being a ruminant, what is the common problem with cattle, yak, yak-cattle hybrids that justifies including them in the study?

Discussion:

Lines 258-281: Enterotyping was initially proposed to differentiate gut microbiota profiles in order to associate them with predispositions or their diet for human. In this study, enterotyping in ruminants appears to be only a possible environment-related grouping. Why is the presence of two enterotypes of interest to the advancement of knowledge for ruminants? This point does not seem to have been clearly specified?

Reviewer #2 (Comments for the Author):

General comments

The authors investigated an interesting studying on the fecal microbiota profiles of domestic ruminants (cattle, yak, yak-cattle hybrid, and Tibetan sheep) dwelling in 4 typical eco-regions with homologous longitudes in Qinghai-Tibetan Plateau Area.

However, the overall quality is relatively weak, especially for the experimental design. There are only 340 samples in this study, but with too many interference factors, such as the sex, age, environment, feed, infections, animal breed, etc.

In addition, the objective of this study is not very clear. If you aim to detect the effect of environment on the gut microbiota in ruminants. The animals and feed should be similar. And what is the meaning or implication of this study that has not been clearly clarified?

There are many grammar and format issues that should be corrected throughout the manuscript. For instance:

Fig. or FIG;

Amplicon Sequence Variants or amplicon sequence variants;

Studies on ruminant microbiota has been already should be "Studies on ruminant microbiota have been..."

Line-by-line comments

Line 23-24: suggest changing to "All of the environment, human activity, species, and parasitization affected the fecal microbiota."

Line 65: suggest changing to "taken into; a big natural; and similar biological"

Line 349-352: suggest using parenthesis rather than square brackets

Figures:

FIG3: What do you mean by the g and f in FIG3b and 3c. Where are the captions of d, e, f, g.

Reference: Please pay more attention to the format of the Reference following the instruction to authors. Such as 19, 40, 41, 46, ...

Staff Comments:

Preparing Revision Guidelines

To submit your modified manuscript, log onto the eJP submission site at <https://spectrum.msubmit.net/cgi-bin/main.plex>. Go to

Author Tasks and click the appropriate manuscript title to begin the revision process. The information that you entered when you first submitted the paper will be displayed. Please update the information as necessary. Here are a few examples of required updates that authors must address:

Please return the manuscript within 60 days; if you cannot complete the modification within this time period, please contact me. If you do not wish to modify the manuscript and prefer to submit it to another journal, please notify me of your decision immediately so that the manuscript may be formally withdrawn from consideration by Microbiology Spectrum.

Dear Editor and Reviewer,

We would like express our gratitude to the Editor and Reviewer for the earnest and instructive comments and suggestions which would help us to improve the quality of the manuscript. Additionally, the text added in the revised manuscript was highlighted in yellow. The modification and response as follow.

Response to Reviewer 1 Comments

Lines 60-69: The context of the study is described but the problem is not. What is the problem that justifies analyzing the fecal microbiota of four ruminants? Is it the presence of parasites, is it based on geographical differences? Without a clearly stated hypothesis or research question, this study is only descriptive although the number of fecal samples is high. Please identify the problem.

RESPONSE: Thank you very much for your suggestion. Sorry for not communicating clearly. The prime objective in this study is to depict the profile of fecal microbiota of ruminants in the QTPA, and then evaluating the dominant factor that shift microbiota structure, among several factors such as diet (environment), genetics and parasitization. This part has been arranged to suit your suggestion.

Lines 70-71: Why did you also study the fecal microbiota of Tibetan sheep? Besides being a ruminant, what is the common problem with cattle, yak, yak-cattle hybrids that justifies including them in the study?

RESPONSE: Thank you very much for pointing it out. As the manuscript initially outlined that cattle, yak and sheep were the main ruminants in the Qinghai-Tibetan Plateau Area (QTPA). We try analyse the fecal microbiota of these main ruminants to reveal the fecal microbiota characteristic of ruminants in the QTPA. Moreover, the results showed similar conclusions about the importance of environment factor on shifting the fecal microbiota either in sheep or in cattle, yak and yak-cattle hybrids.

Lines 258-281: Enterotyping was initially proposed to differentiate gut microbiota profiles in order to associate them with predispositions or their diet for human. In this study, enterotyping in ruminants appears to be only a possible environment-related grouping. Why is the presence of two enterotypes of interest to the advancement of knowledge for ruminants? This point does not seem to have been clearly specified?

RESPONSE: Thanks very much for your advice. High-altitude animals have evolved physiological adaptations to live in extremely conditions of the QTPA, and for animals, enterotype is an instinctive response to environment. This study not only helped determine the enterotype for ruminants in the QTPA, but also reflected the information of the local vegetation and ecology. Related content has been added to line 281-283 (clean revision).

Response to Reviewer 2 Comments

However, the overall quality is relatively weak, especially for the experimental design. There are only 340 samples in this study, but with too many interference factors, such as the sex, age, environment, feed, infections, animal breed, etc.

RESPONSE: Thank you very much for your suggestion. By agreeing that the sex, age, environment, feed, infections, animal breed could affect the fecal microbiota. Here, even with variables (the sex and age), results still showed the environmental

tendency of the fecal microbiota, environment over other factors in shaping the microbiota structure. Furthermore, in this study, sampling, preservation condition, experimental methods and sequencing of all samples were carried out under the same conditions, to reduce the error. As mentioned at the end of the manuscript, this study demonstrated the importance of the environment, but there are still many more samples required to explore and perfect the spectrum of fecal microbiota of ruminants in the QTPA.

In addition, the objective of this study is not very clear. If you aim to detect the effect of environment on the gut microbiota in ruminants. The animals and feed should be similar. And what is the meaning or implication of this study that has not been clearly clarified?

RESPONSE: Thanks very much for your suggestion. Sorry for not communicating clearly. Due to diet is part of environment factors, and diet is acquiesced as environment factor here. The effect of environment (diet) was demonstrated from varied ecotypes in different typical eco-regions, such as yaks from 4 eco-regions and Tibetan sheep from 3 eco-regions, in which shared the same breeds but diverse regions. The genetics or species influence was confirmed by multiple species or varieties (cattle, yak and yak-cattle hybrid) in one habitat (Diqing), which grew under similar environment and forage.

There are many grammar and format issues that should be corrected throughout the manuscript. For instance:

Fig. or FIG;

Amplicon Sequence Variants or amplicon sequence variants;

Studies on ruminant microbiota has been already should be "Studies on ruminant microbiota have been..."

RESPONSE: Thanks so much for your reminding. the above questions have been modified and the grammar has been checked throughout the whole manuscript.

Line-by-line comments

Line 23-24: suggest changing to "All of the environment, human activity, species, and parasitization affected the fecal microbiota."

Line 65: suggest changing to "taken into; a big natural; and similar biological"

Line 349-352: suggest using parenthesis rather than square brackets

RESPONSE: Thank you very much. We have modified the above questions as you suggested.

Figures:

FIG3: What do you mean by the g and f in FIG3b and 3c. Where are the captions of d, e, f, g.

RESPONSE: Thank you very much for pointing it out. Sorry for making a mistake here and we have corrected the error.

Reference: Please pay more attention to the format of the Reference following the instruction to authors. Such as 19, 40, 41, 46, ...

RESPONSE: Thanks so much for alerting us to the problem. The format has been fixed.

May 22, 2022

Prof. Ziyuan Duan
Institute of Genetics and Developmental Biology, Chinese Academy of Sciences
Beijing
China

Re: Spectrum00021-22R1 (Depicting Fecal Microbiota Characteristic in Yak, Cattle, Yak-Cattle Hybrid and Tibetan Sheep in Different Eco-Regions of Qinghai-Tibetan Plateau)

Dear Prof. Ziyuan Duan:

Link Not Available

Sincerely,

Jing Han

Journals Department
Reviewer comments:

Reviewer #2 (Comments for the Author):

Reviewers' comments:

1) However, the overall quality is relatively weak, especially for the experimental design. There are only 340 samples in this study, but with too many interference factors, such as the sex, age, environment, feed, infections, animal breed, etc.
RESPONSE: Thank you very much for your suggestion. By agreeing that the sex, age, environment, feed, infections, animal breed could affect the fecal microbiota. Here, even with variables (the sex and age), results still showed the environmental tendency of the fecal microbiota, environment over other factors in shaping the microbiota structure. Furthermore, in this study, sampling, preservation condition, experimental methods and sequencing of all samples were carried out under the same

conditions, to reduce the error. As mentioned at the end of the manuscript, this study demonstrated the importance of the environment, but there are still many more samples required to explore and perfect the spectrum of fecal microbiota of ruminants in the QTPA.

New Comments: I still suggest that the authors should list out the sex and age, and show their effects in different region.

2) In addition, the objective of this study is not very clear. If you aim to detect the effect of environment on the gut microbiota in ruminants. The animals and feed should be similar. And what is the meaning or implication of this study that has not been clearly clarified?

RESPONSE: Thanks very much for your suggestion. Sorry for not communicating clearly. Due to diet is part of environment factors, and diet is acquiesced as environment factor here. The effect of environment (diet) was demonstrated from varied ecotypes in different typical eco-regions, such as yaks from 4 eco-regions and Tibetan sheep from 3 eco-regions, in which shared the same breeds but diverse regions. The genetics or species influence was confirmed by multiple species or varieties (cattle, yak and yak-cattle hybrid) in one habitat (Diqing), which grew under similar environment and forage.

New Comment: We know that the diet is included into the environment factors, but you can not say that diet is acquiesced as environment. Thus, in the conclusion you mention that the fecal microbiota dissimilarity based on environment rather than host factor among ruminants in the QTPA. The environment should be the most significant factors. What is new from your study? If your objective is to show the difference of fecal microbiota among the four regions, you need list out them in the conclusion, rather than just say the environment rather than host factor. Please make the aim of this study consistent and more clear.

Staff Comments:

Preparing Revision Guidelines

Please return the manuscript within 60 days; if you cannot complete the modification within this time period, please contact me. If you do not wish to modify the manuscript and prefer to submit it to another journal, please notify me of your decision immediately so that the manuscript may be formally withdrawn from consideration by Microbiology Spectrum.

Dear Editor and Reviewer,

We would like express our gratitude to the Editor and Reviewer for the earnest and instructive comments and suggestions which would help us to improve the quality of the manuscript. Additionally, the text added in the revised manuscript was highlighted in yellow. The modification and response as follow.

Response to Reviewer 2 Comments

COMMENTS: I still suggest that the authors should list out the sex and age, and show their effects in different region.

RESPONSE: Thank you very much for your suggestion. The brief information about the age and sex of all samples were listed in Table S1. The PCoA plot of TS-CK (left) and TS-TD (right) as shown below, it was clear that samples are grouped by region, while gender has little effect on the fecal microbiota structure. Although age affects the fecal microbiota, the tendency is not significant in our samples.

As you suggest, the sex and age effects have been shown in the revised manuscript. The related PCoA plots of each group have been added to supplemental file and named as FIG S2. The related description and discussion have been added to line 145-148 and 335-342.

COMMENTS: We know that the diet is included into the environment factors, but you can not say that diet is acquiesced as environment. Thus, in the conclusion you mention that the fecal microbiota dissimilarity based on environment rather than host factor among ruminants in the QTPA. The environment should be the most significant factors. What is new from your study? If your objective is to show the difference of fecal microbiota among the four regions, you need list out them in the conclusion, rather than just say the environment rather than host factor. Please make the aim of this study consistent and more clear.

RESPONSE: Thanks very much for your suggestion. In order to accurately describe this work, the environment has been changed to diet as you suggested. Moreover, the conclusion has been modified.

June 8, 2022

Prof. Ziyuan Duan
Institute of Genetics and Developmental Biology, Chinese Academy of Sciences
Beijing
China

Re: Spectrum00021-22R2 (Depicting Fecal Microbiota Characteristic in Yak, Cattle, Yak-Cattle Hybrid and Tibetan Sheep in Different Eco-Regions of Qinghai-Tibetan Plateau)

Dear Prof. Ziyuan Duan:

Your manuscript has been accepted, and I am forwarding it to the ASM Journals Department for publication. You will be notified when your proofs are ready to be viewed.

Sincerely,

Jing Han
Editor, Microbiology Spectrum

Journals Department
Supplemental file 1: Accept